# Increasing Trends in Obesity-Related Cardiovascular Risk Factors in Romanian Children and Adolescents—Retrospective Study

**DOI:** 10.3390/healthcare10122452

**Published:** 2022-12-05

**Authors:** Alina-Costina Luca, Alexandrina-Stefania Curpan, Elena Emanuela Braha, Elena Ţarcă, Alin-Constantin Iordache, Florin-Alexandru Luca, Heidrun Adumitrachioaiei

**Affiliations:** 1Department of Mother and Child Medicine–Pediatric Cardiology, Faculty of Medicine, Gr. T. Popa’ University of Medicine and Pharmacy, 700115 Iasi, Romania; 2Sfânta Maria’ Emergency Children’s Hospital, 700309 Iasi, Romania; 3Department of Biology, Faculty of Biology, “Alexandru Ioan Cuza” University of Iasi, Bd. Carol I, 20A, 700505 Iasi, Romania; 4National Institute of Endocrinology CI Parhon, Department of Genetics Endocrinology, B–dul Aviatorilor, nr. 34–38, Sector 1, 011863 Bucureşti, Romania; 5Department of Surgery II–Pediatric Surgery, Grigore T. Popa’ University of Medicine and Pharmacy, 700115 Iasi, Romania; 6Department BMTM, “Gheorghe Asachi” Technical University, Bulevardul Profesor Dimitrie Mangeron 67, 700050 Iaşi, Romania

**Keywords:** cardiovascular impairment, obesity, children, echocardiography

## Abstract

Childhood obesity has become a global public health issue and its assessment is essential, as an obese child is a future overweight or obese adult. Obesity is no longer a matter of exercising more and eating less, with several factors coming into play and dictating the pattern of fat accumulation and the ease/difficulty of reducing it. In the current paper, we aimed to analyze the cardiovascular impact of obesity in a large number of patients alongside the paraclinical changes that occur due to weight gain, and to perform an analysis on the increase in prevalence throughout our research. The main cardiovascular conditions identified were hypertension (15.36%), septal or concentric hypertrophic cardiomyopathy (11.15%), atherosclerosis risk (13.04%), and hypercholesterolemia (20.94%). We have used echocardiography to measure the thickness of epicardial adipose tissue (useful for assessing the patient’s cardiovascular risk), and we observed that it was greater in children with moderate and severe obesity alongside diastolic dysfunction of the left ventricle in the whole group, without any connection with hypertension or coronary impairment. Obese children will be affected by increased cardiovascular mortality and morbidity in adulthood and they may experience early cardiovascular dysfunction. We want to strongly underline the importance and necessity of programs for the early detection and prevention of obesity and its complications, especially since interesting phenomena such as the “obesity paradox” exist and prove that obesity is far less understood than it is at a first glance.

## 1. Introduction

In recent years, obesity has started to be seen more and more as an epidemic [1], with a worrying number of individuals reaching the morbidity stage in terms of health risks. One aspect that tends to be overlooked is regarding childhood obesity. Children and adolescents that start to be overweight while they are still in the developing stages both physically and mentally, tend to carry that weight into adulthood with worsening symptoms and, in turn, become obese adults. Therefore, research and studies focusing on overweight children are of utmost importance for our society as weight comes with a series of associated comorbidities who would not be normally found in otherwise healthy children, such as hypertension, cardiovascular threats, metabolic manifestations down to morbidity and premature death. As studies have shown, a higher weight in childhood leads to a higher risk of fatal and nonfatal cardiac events in adults [2,3].

According to the World Health Organization (WHO), in 2016 over 340 million children aged 5 to 19 years old were classified as overweight or obese whereas in 2020, 39 million children younger than 5 years old were considered overweight or obese; the global level has drastically increased since 1975 [4] These numbers are also in accordance with the UNICEF/WHO/World Bank Group Joint Child Malnutrition most recent estimate (2021) [5]. Therefore, 1 in 3 children have weight issues, making childhood obesity one of the most important public health concerns in developed and developing countries [4].

Although there is a global action meant to halt the rates of obesity by 50% by the year 2025 as a part of reducing the range of noncommunicable diseases, the numbers keep rising and affecting more and more countries every year. The current predictions show that by 2030, 1 in 8 of all children and adolescents (aged 5 to 19) will be obese [6,7]. The highest rates of overweight and obesity are in the Americas and in the European region. In Europe, the trends are higher in the Mediterranean and eastern parts and is associated with lower education and incomes, whereas in middle income countries the rates were reversed [8]. One study has found that in lower income countries, 2 in 5 children aged 6 to 9 years old are overweight/obese [9].

The World Obesity Federation estimates that there will be almost 500,000 obese children aged between 5 and 18 in Romania by 2030, unless harsh prevention and treatment measures are implemented [10]. The results of the Health Profile in Romania were presented to the European Commission in 2017, and the results illustrated that the overweight and obesity rates among adolescents have doubled over the last decade, increasing from 7% in 2005–2006 to 16% in 2013–2014 [11].

In the context of the increasing prevalence of obesity among the pediatric population, the imminent danger of increasing incidence for cardiovascular diseases is also discussed, especially since most often an obese child is a future obese adult, as a high BMI during adolescence is predictive of a higher cardiovascular disease risk and a higher mortality rate in adulthood [12]. Therefore, prevention methodologies against the increase in children and adolescents affected by this type of malnutrition have become of utmost importance for reducing chronic diseases in adults [13].

Obesity is no longer a matter of exercising more and eating less, with several factors coming into play and dictating the pattern of fat accumulation and ease/difficulty to reduce it. Such factors include specific phenotypes [14], social ties and peer pressure [15], food cues [16], education [17] and income [18] level as well as disorders such as polycystic ovary syndrome [19] which affects the female ability to lose or gain weight which in turn can influence the child before it is born. Amongst all risk factors triggered by obesity, the cardiovascular ones are the hardest to diminish as a higher BMI and waist are clearly associated with cardiovascular events and type 2 diabetes mellitus. CVDs appear at a younger age and with more comorbidities in obese individuals contrary to their non-obese counterparts [14].

As simple as it might seem, obesity is in fact a disorder with a multifactorial etiology as its occurrence requires multiple interactions between genetic, behavioral, psychological, and metabolic factors, which in turn result in energy balance changes [20]. The WHO defines overweight and obesity as excessive or abnormal accumulations of fat that can affect the health. Obesity is a public health concern since affected patients need long-term monitoring and treatment of related conditions. A body mass index (BMI = kg/m^2^) between the 85th and 95th percentiles is classified as overweight, whereas a BMI over the 95th percentile as obesity. Children/adolescent’s obesity is defined and classified using age- and sex-specific growth charts. Large scientific studies have demonstrated impacts on all major organ systems of the body, especially related to future coronary heart disease risk in adult life [21]. However, there is a phenomena known as the “obesity paradox” according to which overweight and obese individuals have better prognosis and CVD improvements compared to the leaner individuals [22,23,24,25] which suggest that obesity is nowhere near well-understood and further studies can help elucidate some of its mysteries and help establishing better therapeutic ways to improve and protect the health of individuals.

Our research aims to assess the trends of the cardiovascular risk profiles associated with obesity in patients admitted to the Cardiology Clinic of ‘Sfanta Maria’ Children’s Hospital of Iasi over a period of 14 years; to the best of our knowledge, there is no other study focusing on overweight and obese pediatric patients over a long period of time. Given the massive increase in obesity among pediatric patients, the wide range of high-calorie foods accessible to all, lack of exercise and patients’ and often their family’s lifestyle, our research seeks to identify clinical, causal, and demographic patterns to facilitate diagnosis setting, treatment and design of effective prevention systems.

## 2. Materials and Methods

In order to reach our research goals, we conducted a retrospective descriptive study conducted over a 14-year period, based on the total number of patients hospitalized in the Cardiology Department of ‘Sfanta Maria’ Children’s Hospital of Iasi.

Therefore, 1165 children, included in the study, were hospitalized here between 1st January 2006 and 1st January 2020 (14 years). The inclusion criterion consisted of a body mass index ranging between the 85th and 95th percentiles for overweight and exceeding the 95th percentile for obesity. The standard period of hospitalization in the Pediatric Cardiology Department was 7 days for a complete run of clinical and paraclinical examinations. However, if the patients were suspected of rhythm disorders, the hospitalization period was extended with 24 h for monitoring with a Holter device in order to confirm or rule out that diagnosis.

The research protocol consisted of the analysis of all patients’ records and the extraction of data related to their clinical examination at the time of the echocardiographic evaluation-assessment of race, sex, age, height and weight development by determining weight, height and calculation of body mass index; its value being interpreted according to the patients’ age and gender using special growth charts provided by the WHO in 2006 and 2007, and recommended by the European Society of Gastroenterology, Hepatology and Pediatrics Nutrition (ESOGHAN).

Clinical quantification of obesity in children over 2 years of age and adolescents was carried out using the body mass index (BMI) calculated according to the known formula (BMI = kg/m^2^) and interpreted using percentiles dependent on age and gender. Waist and hip circumferences were obtained by using a measuring tape following the norms already described [26].

The detailed physical exam included, in addition to anthropometric data, skin examination (acanthosis nigricans, hirsutism, striae). For assessing the cardiovascular system health, we performed physical exam, electrocardiography, and echocardiographic exams. The protocols have been maintained throughout the whole study.

All echocardiographic images were collected with the subject in the left decubitus position to achieve parasternal long- and short-axis images.

Medical imaging examinations consisted of echocardiography scans performed on the whole group and designed to determine the ejection fraction of the left ventricle, the thickness of the interventricular septum, the mass of the left ventricle, the diastolic diameter and the posterior wall of the left ventricle, the presence of epicardial fat, and the identification of congenital heart malformations, which may be part of genetic syndromes; electrocardiography, abdominal ultrasound scanning, and thyroid ultrasound scanning.

Subjects known at the time of echocardiographic examination to have congenital cardiac anomalies were excluded from the study.

We observed an association between increased body mass and left ventricular mass-z score with mean z-score being higher in overweight and obese children. Left ventricular mass z score or left ventricular hypertrophy (LVH) was defined as >95th percentile dependent on age [27].

Epicardial adipose tissue was proposed to be a cardiovascular risk predictor [28] and was noted to be correlated with visceral adipose tissue deposition [29].

Blood Pressure was measured and classified by Fourth Report on the Diagnosis Evaluation, and Treatment of High Blood Pressure in Children and Adolescents, with updates from 2017 [30] and it was determined after the child had been lying down for at least 5 min, with the 2 measurements made every 10 min.

Hypertension was determined based on blood pressure value curves (50th, 90th, 95th and 99th percentiles), taking into account the patient’s height (expressed in percentiles), gender and age; the systolic and diastolic blood pressure values under the 90th percentile for each gender, age and height were noted as normal blood pressure; prehypertension was diagnosed when the mean systolic and diastolic blood pressure value ranged between the 90th and <97.5th percentiles for each age, gender and height; it should be noted that in adolescents prehypertension is diagnosed based on the values used in adults: systolic blood pressure > 120 mmHg and diastolic blood pressure > 80 mmHg, while the diagnosis of hypertension was set when the value of systolic and diastolic blood pressure was higher than the 97.5th percentile for each gender, age and height.

Supplementary exams were carried out to assess the state of atherosclerosis-promoting risk factors (dyslipidemia, hypertension, insulin resistance, and smoke exposure). We evaluated the carbohydrate metabolism by determining fasting blood glucose levels, HbA1c and OGTT. We analyzed liver function by dosing total serum proteins, aspartate transaminase(AST) and alanine transaminase(ALT) and uric acid. We determined thyroid hormones, inflammatory markers: rate of sedimentation of hemorrhoids, plasma fibrinogen and C-Reactive Protein.

We analyzed the following data for the whole group: endocrinological exam, nutritional assessment, psychological and neuropsychiatric exam.

The independent variables of the study were gender, origin and genetic predisposition, while the dependent variables were diet, related conditions and physical exercise.

## 3. Results

The research was conducted on 1165 cases of overweight or obese children out of the 20,604 pediatric patients hospitalized in the Cardiology Department of ‘Sfanta Maria’ Children’s Hospital of Iasi over the past 14 years (1 January 2006 to 1 January 2020). The graphical representation of the studied group can be visualized in Figure 1.

The analysis of the group by years indicates an increase in the number of patients diagnosed with obesity since 2014. In other words, 860 of the 1165 obese children were admitted to our clinic in the last 6 years of the study, which means 73.81% of the whole studied group. This increase in the number of cases in the last 6 years overlaps with the new data announced by the WHO, which has warned that 50% of the world’s population will be obese by 2025 unless taken serious measures.

The analysis of the patients by age groups shows a higher percentage (77.93%) in the 10+ age group. All results are summarized in Table 1.

When analyzing distribution by gender, we found almost equal numbers of males (51.08%) and females (48.92%). The trend of gender equality has also been noticed in the statistical data of recent years. There were more patients diagnosed with obesity living in rural areas, i.e., 55.96% of the whole studied group, which is in accordance with the European reports according to which there are higher obesity rates in poorer areas with parents that have a lower education [8]. As their access to information, and medical and psychological counseling is more limited, rural children will more often grow up to become obese adults [8].

The echocardiography revealed that the whole group suffered from diastolic dysfunction of the left ventricle, followed by ventricular hypertrophy (21.8%), and hypertrophic cardiomyopathy (11.15%).

We measured epicardial fat thickness in all patients and found higher values in those with moderate and severe obesity (64.97%) and in those included in the 10+ age group (77.93%). In line with other studies, we noticed an increase in body weight with age, which affected female patients more.

In our research group, 15.36% had hypertension, and 47.48% of these needed to medication together with diet and exercise.

After analyzing inflammatory markers, we noticed a risk of atherosclerosis (13.04%) with high fibrinogen, 5.92% of whom also suffered from an inflammatory syndrome, while the other 6.57% only had high fibrinogen levels and no inflammatory syndrome.

It is already common knowledge that the genetic factors acting in combination with environmental factors and the anamnesis that we performed revealed that 16.99% of the hospitalized patients had a family history of obesity. Thus, according to WHO data, diet mistakes associated with lack of exercise make one in four 11-year-olds overweight or obese.

An important aspect is the fact that psychological examination revealed that 15.96% of them suffered from mild depression and low self-esteem, while 11.93% had suicidal tendencies blamable on their physical appearance and difficult social integration.

## 4. Discussions

Overweight and obesity are significant risk factors for the occurrence of negative cardiovascular manifestations, they are closely related to endothelial dysfunction, abnormal geometry of the left ventricle, left ventricular systolic and diastolic dysfunction, increased arterial stiffness, left atrium dilation, atrial fibrillation and generally heart failure [31].

The cardiovascular system is subject to significant structural and functional physio-pathological changes in obesity: cardiomyocyte fibrosis and apoptosis, mitochondrial dysfunction, extracellular matrix changes, adipokine production disorder [32,33].

Pressure and volume overload, with one of them being predominant, is a mixed mechanism of left ventricular overload that characterizes obesity depending on which left ventricular (concentric or eccentric) remodeling occurs [34]. Left ventricular remodeling was also present in the patients included in our research group.

Echocardiography showed that diastolic dysfunction of the left ventricle was present in the whole group, while hypertrophic cardiomyopathy affected 11.15% and hypertrophy of the left ventricle 21.8% of them.

In pediatrics, epicardial tissue thickness, which is considered an organ with many functions—metabolic, thermoregulatory and mechanical—and also considered by researchers as a predictor of cardiovascular risk, should be 4.1 mm [35,36].

Epicardial fat measurement was part of the echocardiographic examination performed. Just as in other studies, the patients in our group had more epicardial fat, which was directly proportional to their weight gain and age.

In a study of 341 Japanese obese children aged from 6 to 15, echocardiography determined the thickness of the interventricular septum, the thickness of the posterior wall of the left ventricle, the diastolic diameter and the mass of the left ventricle. It revealed that left ventricle changes began to occur in obese children as early as the age of 6 [37,38].

In another study conducted on a smaller number of children (45), which also included normal body weight children, echocardiography was performed in both groups and revealed that left ventricle mass was greater in obese children, yet the degree of left ventricular hypertrophy was not correlated with the degree of obesity [39].

The prevalence of hypertension is 3–5% in childhood; from puberty an increase in prevalence is observed and from 18 years the prevalence is similar with adulthood prevalence (10–11%) [40,41,42,43,44,45].

In our study, 15.36% of the patients suffered from hypertension, and 47.48% of them needed medication, diet and exercise.

The relationship between BMI and blood pressure values was observed in a big study in countries such as China, Korea, Poland, USA, India, Iran and Tunisia. The results of the study show direct relationship between hypertension and obesity after an BMI higher than 25 [40,46].

The analysis of the inflammatory markers data revealed an atherosclerosis risk in 13.04% of them, 5.92% of whom also had inflammatory syndrome with higher fibrinogen levels. Studies have shown that subacute inflammatory processes in blood vessels characterized by infiltration of macrophages and T cells that interact with each other and together with the arterial wall is a physio-pathological process similar to the inflammatory process in the blood vessels in an obese patient. Therefore, the similarities discovered between the two inflammatory processes are currently considered innovative and are integrated in the prevention, diagnosis and treatment of these two diseases [47].

A WHO-UNICEF study conducted in Europe in 2011 revealed approximately 6.9 million children under the age of 5 who were overweight or obese. This is the double of the 1990 number, when 3.3 million obese or overweight children were diagnosed in this age group [48]. The sharp increase in the number of cases diagnosed in the last 6 years of our study worries us and supports the data found in all WHO studies and reports of recent years, especially in the Global Atlas on Childhood Obesity, where the number of overweight children is expected to increase by 100 million between 2020 and 2030, and 20% of children and adolescents affected by obesity are expected to suffer from severe obesity by 2030, requiring specialized healthcare, including surgery [10]. Longitudinal studies show that 50 to 75% of obese adolescents will become obese adults, thus increasing the number of individuals diagnosed with cardiovascular diseases [49].

Recent data from the COSIWHO (Childhood Obesity Surveillance Initiative) report, which also refers to Romania, shows that the distribution by gender is approximately the same, a trend of equalization visible in other studies as well, with 48.92% girls and 51.08% boys in our study [50].

For a long time, studies suggested that urbanization was an important factor in increasing obesity rates worldwide. A new large-scale study conducted by researchers at the Imperial College of London, UK, and published in the journal *Nature*, analyzing over 112 million adults in 200 countries over a 32-year period, i.e., between 1985 and 2017, reveals that things have changed. Thus, urban inhabitants have access to better education, more complex information, higher income which allows them to purchase healthy food and many exercise facilities, while rural inhabitants are lacking some of these opportunities. Moreover, the modernization of rural areas has led to less physical exertion, as physical work has been largely replaced by professional automatic equipment, and the ease of access to unprocessed food together with their lower income have led to higher percentages and to a reverse in the previously known perception. This novelty is also visible in our research, where 55.96% of the 1165 patients lived in rural areas [51].

The ideal treatment for this category of patients is proper diet, with the reduction in the intake of calories by decreasing carbohydrates and lipids yet assuring the amount of protein required for proper and healthy growth and development. When it comes to the standard of healthy patients, the protein intake plays a major role by avoiding a negative protein balance. For the best results, in addition to changing eating habits, children should exercise and receive behavioral counseling. By including them to become part of their lifestyle, the chances of good long-term results are significantly higher [52,53].

A hypocaloric and hypoglycemic diet is the main method to control juvenile obesity, the right proportion being 55% carbohydrates, 15% protein and 30% lipids. Hypoglycemic diets are not recommended because they may give rise to gallstones [52,54].

Exercise has a positive influence on the cardiovascular system by lowering blood pressure, increasing the diameter of coronary arteries, improving oxygen uptake by the myocardium, lowering triglycerides, increasing HDL-cholesterol. The physical activity increases insulin sensitivity, lowers blood sugar levels, favors vascular endothelium’s function, lowers arterial stiffness and blood pressure. Regular exercise is recommended, with increasing duration and intensity of effort [55,56,57,58,59,60].

Medication is rarely recommended, namely when the proper diet approach has been unsuccessful, despite family compliance, behavioral therapy and exercise. It is contraindicated in children under 14-15 years of age, and it does not replace a healthy lifestyle [52,61,62,63,64]. Cellulose-type medication, that increases the volume of gastric contents, is recommended; it is a medication which makes the patient feel hungry less frequently. Gastrofibran is one example [65]. Orlistat, a lipostatin derivative, is also used with severe obesity and significant metabolic alterations [66,67,68]. Bariatric surgery is indicated when the BMI is higher than 40 kg/m^2^, in morbid obesity or in the case of related diseases [69,70].

With the help of our nutritionist and psychologist, all of the patients in our research group were advised to change their lifestyle, enjoyed a customized diet according to their individual BMI, related conditions and family willingness to comply with this change. Where appropriate, cardiological, nutritional and psychological reassessments were performed on all our patients. Family support is vital, as overweight or obese children and adolescents need their family’s support to successfully lose and then maintain normal body weight [71,72,73].

Mental health manifestations include low self-esteem, distorted self-image and depression, which may result in suicidal tendencies, according to the Global Evolution of Obesity Research in Children and Youths: Setting Priorities for Interventions and Policies. There are relatively few studies showing the effects of obesity on the psyche. The same study points out that obesity and depression are the two main public health issues in the pediatric population [74]. In our study, psychological examination revealed that 15.96% of the patients suffered from mild depression and low self-esteem, while 11.93% had suicidal tendencies blamable on their physical appearance and difficult social integration.

Some limitations of our study include the lack of follow-up information on whether or not the cardiovascular risk factors have decreased upon weight loss and the corresponding statistical data. Future studies should also focus on the environmental risk factors leading up to overweight and obese diagnosis, the preventable causes and improvements of the way it is managed once the patient is discharged.

## 5. Conclusions

Obesity has become one of the biggest problems of the 21st century, which affects individuals regardless of gender, age or race, and which requires a unified approach, consisting of the promotion of a healthy lifestyle and healthy eating habits; this includes pregnant women even before conception, who should continue with exclusive natural feeding during the first 6 months of life and end with a diversified and balanced diet, and regular exercise.

High body mass index in adolescence is associated with higher cardiovascular disease risks and a high mortality rate at an early age. Thus, the risk of suffering from hypertension is 3 times higher in obese children than in normal weight children. Pediatric hypertension predisposes to hypertension in adulthood.

The high percentage of patients aged 10 to 18 years in our group should raise the alarm, as it shows the need for obesity screening programs in adolescents and for measures to prevent weight gain as early as possible. As it is difficult to impose dietary restrictions on adolescents, the failure rate and the percentage of obese adolescents who become obese adults are high.

Preventive measures will reduce the number of individuals diagnosed with obesity and hence the number of individuals who will suffer from heart diseases in the future, while reducing the costs of hospitalization and therapy of people diagnosed with cardiovascular diseases.

## Figures and Tables

**Figure 1 healthcare-10-02452-f001:**
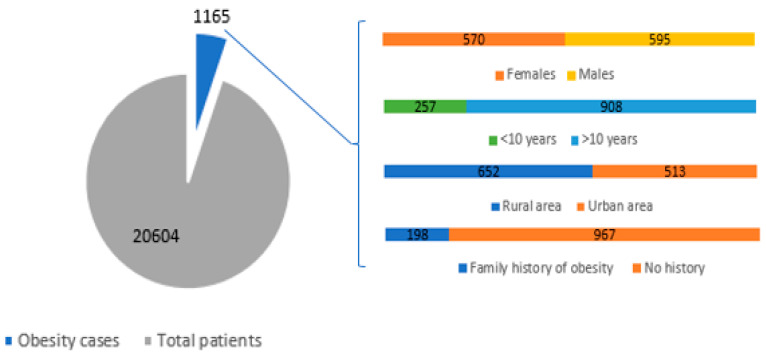
Graphical representation of the studied group.

**Table 1 healthcare-10-02452-t001:** Results of our study summarized.

*Characteristics*	*Obesity/Total Pediatric Patients*	*Proportion*
**Diagnosed with obesity**	1165/20,604	5.65% of all cases
**Age groups**		
<10 years	257	22.07%
>10 years	908	77.93%
**Sex**		
Female	570	48.92%
Male	595	51.08%
**Living area**		
Rural area	652	55.96%
Urban area	513	44.04%
Family history of obesity	198	16.99%
**Clinical features**		
Diastolic dysfunction	1165	100%
Ventricular hypertrophy	254	21.8%
Hypertrophic cardiomyopathy	130	11.15%
**Epicardial fat thickness**		
In Moderate and severe obesity	757	64.97%
In the >10 old group	908	77.93%
**Hypertension**	179	15.36%
With need of medication, diet and exercise	85	47.48% of 15.36%
**Risk of atherosclerosis** with high fibrinogen	152	13.04%
And inflammatory syndrome	9	0.772%
No inflammatory syndrome	10	0.858%
**Hypercholesterolemia**	244	20.94%
Mild depression and low self-esteem	186	15.96%
Suicidal tendencies related to physical appearance and difficult social integration	139	11.93%

## Data Availability

All the data are available in the present paper.

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
