# Peer review of "Increasing Trends in Obesity-Related Cardiovascular Risk Factors in Romanian Children and Adolescents—Retrospective Study"

_healthcare, 2022, doi:10.3390/healthcare10122452_

Round 1
Reviewer 1 Report
Authors have determined different risk factors for CVD in obese children and found significant relationships with obesity and cardiatic functions. Overall, the study is well designed and prestented. I would like the auhtors to address the following:
1. Line 51 provides WHO obesity stats from 2013. Is not there any more recent data for this? They also state that more than 42 million preschollers world wide to be obese. Please check these numbers it may be more than what is listed.
2. L114 states 1165 chldren were hospitalized. Please state for how many days.
3. Line 133 He protocol-------------. What do you mean by He?
4. Line 155, please specify what was examined?
5. Line 305, exercise does not increase IR as stated. It imporves insulin senstivity and decreases IR.
6. Lines 153-157, CRP, HbA1c, OGGT, CRP were monitored but no data are shown. It will be interest to the reader.
7. Table 1 should include inforamtion re high BP, high chol, Inflamatory syndrome
Author Response
Dear Reviewer 1,
Thank you for your time and useful observations. We have chose to respond as follows:
- Line 51 provides WHO obesity stats from 2013. Is not there any more recent data for this? They also state that more than 42 million preschollers world wide to be obese. Please check these numbers it may be more than what is listed.
We apologize for the incorrect data, we have verified the information and corrected with the correct numbers. Besides who, UNICEF/WHO/World Bank Group Joint Child Malnutrition most recent estimate (2021) predicted an estimate of 38.9 million of overweight and obese children under 5 by 2020 which is in accordance with the data presented (Lines 51-56).
2. L114 states 1165 chldren were hospitalized. Please state for how many days.
Thank you for your observation. The children were hospitalized for a standard period of 7 days with a 24h extension if the patient was suspected of rhythm disorders (Lines 121-125).
3. Line 133 He protocol-------------. What do you mean by He?
Thank you for pointing it out, it was just a writing mistake. We have corrected it to "The".
4. Line 155, please specify what was examined?
We analyzed liver function by measuring alanine transaminase (ALT) and aspartate transaminase (AST), total serum proteins, fibrinogen and uric acid (Lines 163-164).
5. Line 305, exercise does not increase IR as stated. It imporves insulin senstivity and decreases IR.
Thank you for your observation, we have corrected it with "The physical activity increases insulin sensitivity, lowers blood sugar levels, favors vascular endothelium’s function, lowers arterial stiffness and blood pressure." (Lines 314-315).
6. Lines 153-157, CRP, HbA1c, OGGT, CRP were monitored but no data are shown. It will be interest to the reader.
Whereas it is true that we have not included the exact value, we considered more of interest the prevalence of certain high-risk disorders with our group of study as our analysis focuses on the connections between overweight and obesity to cardiovascular risk factors.
7. Table 1 should include inforamtion re high BP, high chol, Inflamatory syndrome
The Table does contain information about the prevalence of hypertension, atherosclerosis, inflammatory syndrome and hypercholesterolemia amongts our group of study.
Thank you for your observations.
Reviewer 2 Report
Luca and co-authors has reported a mini review and retrospective study on obesity-related cardiovascular risk factors in Romanian children and adolescents. The authors analyzed the cardiovascular impact of obesity in a large number of patients alongside the paraclinical changes that occur due to weight gain. They also performed an analysis on the prevalence increase throughout the research. Childhood obesity has been overlooked and this manuscript reported some interesting findings however, I have some concerns about the present manuscript.
1. The title of this manuscript was mini review and retrospective study. The literature review in this manuscript was comparable to a standard research paper. The authors should omit the word “mini review” in the title unless they could do a more detail literature review.
2. Line 144-145, the description on the left ventricular mass Z score was incomplete.
3. Line 161, the term “genetic component” should be more specific.
4. The retrospective study was a description of clinical data without analysis and interpretation. Not much new information was presented. The comparison was not done correctly. The authors had used “significantly” several times, e.g. in line 168, 171, 174. Avoid using “significant” unless the data were compared using statistical tests appropriately.
5. The authors stated that the significant increase in the number of cases overlaps with the new data announced by the WHO which has warned that 50% of the world’s population will be obese by 2025 unless taken serious measures. To me, there was no direct correlation between the data reported here and the prediction the WHO.
6. Line 178-179 duplicated Line 174-175.
7. Line 181-182, a reference was missing.
8. Line 184, a reference was missing.
9. Line 191-192, data presentation was not clear. The authors stated that they measured epicardial fat thickness in all patients and found higher values in those with moderate and severe obesity (64.97%) and those in the 10+ age group (77.93%). According to Table 1, 64.97% and 77.93% represent the proportion. The figures and the text was not consistent.
10. Line 195-205 should be put under “Methods”.
11. Data presentation in Table 1 should be revised. Under “Risk of atherosclerosis”, the authors presented the data as % of % which made the data looks very complicated.
12. Minor, line 133, The protocols
13. Minor, line 155. A missing word before “Was analyzed…”
14. Limitations of the study should be discussed, especially this is a retrospective study.
Author Response
Dear Reviewer 2,
Thank you for your time and for your observations. We have responded as follows:
- The title of this manuscript was mini review and retrospective study. The literature review in this manuscript was comparable to a standard research paper. The authors should omit the word “mini review” in the title unless they could do a more detail literature review.
We have eliminated the mini-review from the title.
- Line 144-145, the description on the left ventricular mass Z score was incomplete.
We have added that there was an association observed between increased body mass and left ventricular mass-z score with the mean being higher in overweight and obese children (Lines 152-153).
- Line 161, the term “genetic component” should be more specific.
We have replaced „genetic component” with „genetic predispozition”.
- The retrospective study was a description of clinical data without analysis and interpretation. Not much new information was presented. The comparison was not done correctly. The authors had used “significantly” several times, e.g. in line 168, 171, 174. Avoid using “significant” unless the data were compared using statistical tests appropriately
We have removed the use of the word significant in relation to our data, however we consider that the information is of interest as it illustrated the increased number of overweight and obese children alongside the increased prevalence of hypertension, atherosclerosis, inflammatory syndrome and hypercholesterolemia.
- The authors stated that the significant increase in the number of cases overlaps with the new data announced by the WHO which has warned that 50% of the world’s population will be obese by 2025 unless taken serious measures. To me, there was no direct correlation between the data reported here and the prediction the WHO.
We consider there is a correlation between our data and the WHO prediction because as stated our study analysed admitted patients over a 14 years period and more than half (860 out of 1165) were admitted in the last 6 years included here which is a worrisome trend.
- Line 178-179 duplicated Line 174-175.
Removed.
- Line 181-182, a reference was missing.
- Line 184, a reference was missing.
Added.
- Line 191-192, data presentation was not clear. The authors stated that they measured epicardial fat thickness in all patients and found higher values in those with moderate and severe obesity (64.97%) and those in the 10+ age group (77.93%). According to Table 1, 64.97% and 77.93% represent the proportion. The figures and the text was not consistent.
It is proportion as 64,97% refers to the number of all ages children diagnosed with moderate and severe obesity and 77,93% represents only the 10 years group that included overweight and obese.
- Line 195-205 should be put under “Methods”.
We moved the specified paragraph to the methods section.
- Data presentation in Table 1 should be revised. Under “Risk of atherosclerosis”, the authors presented the data as % of % which made the data looks very complicated.
We have modified the Table and reported the percentages to the total number of cases included.
- Minor, line 133, The protocols
Corrected
- Minor, line 155. A missing word before “Was analyzed…”
Corrected.
- Limitations of the study should be discussed, especially this is a retrospective study.
Added.
Thank your for your useful remarks.
Round 2
Reviewer 2 Report
The responses from the authors and the revised version of the manuscript looks fine with me.